# Seroprevalence of Antibodies against Tick-Borne Pathogens in Czech Patients with Suspected Post-Treatment Lyme Disease Syndrome

**DOI:** 10.3390/microorganisms9112217

**Published:** 2021-10-25

**Authors:** Kristyna Sloupenska, Jana Dolezilkova, Barbora Koubkova, Beata Hutyrova, Mojmir Racansky, Pavel Horak, Maryna Golovchenko, Milan Raska, Natalie Rudenko, Michal Krupka

**Affiliations:** 1Department of Immunology, Faculty of Medicine and Dentistry, Palacky University Olomouc, Hnevotinska 3, 779 00 Olomouc, Czech Republic; sloupenska.k@gmail.com (K.S.); beata.hutyrova@fnol.cz (B.H.); mojmir.racansky@gmail.com (M.R.); milan.raska@upol.cz (M.R.); 2Laboratory of Medical Parasitology and Zoology, Public Health Institute Ostrava, Partyzanske Namesti 7, 702 00 Ostrava, Czech Republic; Jana.Dolezilkova@zuova.cz; 3Department of Allergology and Clinical Immunology, University Hospital Olomouc, I.P. Pavlova 185/6, 779 00 Olomouc, Czech Republic; barbora.koubkova@fnol.cz; 4Third Department of Internal Medicine-Nephrology, Rheumatology and Endocrinology, University Hospital Olomouc, I.P. Pavlova 185/6, 779 00 Olomouc, Czech Republic; pavel.horak@fnol.cz; 5Institute of Parasitology, Biology Centre, Czech Academy of Sciences, Branisovska 1160/31, 370 05 Ceske Budejovice, Czech Republic; marina@paru.cas.cz

**Keywords:** ticks, co-infection, Lyme disease, post-treatment Lyme disease syndrome, *Bartonella*, *Anaplasma*, *Babesia*, seroprevalence

## Abstract

The hypothesized importance of coinfections in the pathogenesis of post-treatment Lyme disease syndrome (PTLDS) leads to the use of combined, ongoing antimicrobial treatment in many cases despite the absence of symptoms typical of the presence of infection with specific pathogens. Serum samples from 103 patients with suspected post-treatment Lyme disease syndrome were tested for the presence of antibodies to the major tick-borne pathogens *Anaplasma phagocytophilum*, *Bartonella henselae*/*Bartonella quinatana*, and *Babesia microti*. Although the presence of anti-*Anaplasma* antibodies was detected in 12.6% of the samples and anti-*Bartonella* antibodies in 9.7% of the samples, the presence of antibodies against both pathogens in the same samples or anti-*Babesia* antibodies in the selected group of patients could not be confirmed. However, we were able to detect autoantibodies, mostly antinuclear, in 11.6% of the patients studied. Our results are in good agreement with previously published studies showing the presence of a wide spectrum of autoantibodies in some patients with complicated forms of Lyme disease and post-treatment Lyme disease syndrome, but they do not reveal a significant influence of co-infections on the development of PTLDS in the studied group of patients.

## 1. Introduction

Lyme disease is the most common tick-borne infectious disease of humans in both North America and Eurasia. The number of diagnosed and treated cases has reached nearly 480,000 per year in the United States [1] and up to 850,000 new cases per year in Europe [2,3] in the last 10 years. It is a multisystem infectious disease with a diverse spectrum of clinical manifestations caused by spirochetes of the *Borrelia burgdorferi* sensu lato (s. l.) complex, which includes 22 well-established species [4,5,6]. Ten species of this complex have already been detected or isolated in humans [2,7,8]. The disease has an anthropozoonosis character, in which wild animals are the reservoirs and ticks of the genus *Ixodes* transmit the pathogen to humans. The list of animal hosts for ixodid ticks that serve as reservoirs for *Borrelia* currently includes several hundred vertebrate species, including mammals, reptiles, and birds [9].

Lyme disease occurs in three stages—early localized infection, early disseminated infection, and late infection. However, not all stages develop in every infected person. The clinical manifestations most often affect the skin, joints, nervous system and heart. The appearance of erythema migrans, a highly variable skin rash, is the most common early symptom of Lyme disease. According to the CDC, erythema migrans occurs in 70–80% of infected individuals [10]. Although erythema migrans occurs primarily at the site of the tick bite, it can occur anywhere on the body and can expand in size and number. Within days to weeks, *Borrelia* can spread to secondary sites of infection where it can affect the peripheral and/or central nervous system, heart, or joints. In late stages of infection, arthritis of the large joints or skin lesions in the form of acrodermatitis chronica atrophicans may occur [4,11]. *Borrelia* in the early stages of infection are sensitive to a variety of antibiotics, and successful therapy is mainly ensured by substances from the group of tetracyclines (doxycycline) and beta-lactams (penicillin and cephalosporins). There is an ongoing medical debate about the duration of antibiotic treatment of LD patients and the effectiveness of repeated treatment of these patients and patients suffering from PTLDS. Although long-term antibiotic therapy, repeated antibiotic treatments, or a combination of antimicrobial therapies are not currently recommended by most official health authorities [12], a number of published studies support the benefit or efficacy of prolonged antibiotic treatment (for a review, see [13,14]). Although significant improvements were noted after prolonged treatment, relapse of LD symptoms was observed by several authors after the discontinuation of medication [15]. The controversy about the treatment strategy cannot easily be resolved because Lyme disease is a complex condition that requires a complex solution. Possible explanations for the impairment of the results of prolonged or repeated therapy of patients could be found in multiple etiologies of disease, including persistent spirochete infection or other multiple causes leading to an overlap of sources of inflammation (for a review, see [13]). Recently, the possibility of the presence of persistent coinfections with pathogens commonly transmitted by ixodid ticks together with LD spirochetes, especially coinfections with *Babesia* or *Bartonella*, has received increasing attention [16,17]. However, it should also be taken into account that many antibiotics have significant immunomodulatory properties; therefore, their transient effect may be due to the suppression of the pathological reactivity of the patient’s own immune system. Long-term antibiotic therapy also carries significant risks, from the spread of resistant strains to life-threatening infections such as membranous enterocolitis.

Preventive vaccination is not possible because of the lack of a reliable vaccine, the production of which is difficult due to the high genetic diversity of the *B. burgdorferi* sensu lato complex in general and the species pathogenic to humans in particular [18]. Therefore, the best prevention against infection remains to minimize the risk of tick bites (by wearing appropriate clothing with long sleeves and trousers and using repellents during outdoor activities) and to quickly remove ticks that have already attached. Early recognition of symptoms and the use of appropriate antibiotics are critical for successful treatment.

Although antibiotic therapy is effective in most cases, some patients experience long-lasting problems such as myalgia, arthralgia, chronic fatigue, headaches, or cognitive deficits. Various studies report a frequency of these chronic complications in a wide range of 0–40.8% of patients treated with antibiotics, but the most common estimate is between 10% and 15% [19,20]. The persistence of some forms of *Borrelia* or their remnants is a topic of debate due to their still-unclear influence on the pathogenesis of this disease [21,22]. The role of immunopathological reactions triggered by infection is widely recognized, but their triggers and precise mechanisms are still unknown. The use of the terms “chronic Lyme disease” or “post-treatment Lyme disease syndrome” is intensely debated, and as with any controversial topic, each term has its own proponents [23] or opponents [24]. Notwithstanding some key differences, both sides agree that “Chronic Lyme disease (CLD) is a poorly defined term describing the attribution of various atypical syndromes to protracted *Borrelia burgdorferi*” [24] or “Chronic Lyme disease has been a poorly defined term and is often dismissed as a fictitious entity” [23]. In this paper, we have not attempted to evaluate the lack of proper terminology or to discuss the limitations of the pre-existing entity. The rapidly evolving scientific and clinical understanding of the persistent manifestations of LD in many patients over time and the advances in the complex research on this topic will lead to the development of better diagnostics and hopefully the eradication of this disease.

The diagnostic criteria for Lyme disease are a major problem worldwide. The diagnosis of PTLDS is challenging because symptoms often mimic those of a variety of known diseases and are mainly based on subjectively perceived symptoms such as pain, fatigue, or cognitive impairment [12,25]. Specific markers for persistent syndrome are still missing, and PTLDS is actually a diagnosis of exclusion that does not provide definitive certainty.

The involvement of other tick-borne pathogens and their potential impacts on PTLDS have been widely discussed. Pathogens thought to accompany Lyme disease spirochetes in ticks include a wide range of bacteria (*Anaplasma* spp., *Bartonella* spp., relapsing fever spirochetes, *Rickettsia* spp.), parasites (*Babesia* spp.), or viruses (tick-borne encephalitis virus, Powassan virus) [13,26]. The most commonly discussed types of microorganisms are the bacteria *Anaplasma phagocytophilum* and *Bartonella henselae* and the parasitic piroplasma *Babesia microti*. The mechanism by which coinfections contribute to the pathogenesis of long-term postinfectious health problems is not known. Although most coinfections are subclinical or cause milder manifestations in healthy younger individuals, they can have quite different effects in immunocompromised or elderly patients. When LD and frequent coinfections cannot be treated with currently used drugs, it could be due to the emergence of persistent or dormant forms of spirochetes [21] or drug-resistant concomitant bacteria or parasites, or metabolic differences between pathogens [27]. The need to develop new, more effective LD or PTLDS treatment protocols directed on new drug targets or involving newer drug regimens or therapeutic strategies is clear and must target the primary cause of LD and possible tick-borne co-infections, as well as alternative/persistent/dormant forms of pathogens, including biofilms [13].

An in vitro study has shown that coinfection of *B. burgdorferi* and *A. phagocytophilum* leads to increased production of matrix metalloproteases, cytokines, and/or chemokines in the culture of human brain microvascular endothelial cells [28]. This indirectly suggests that the pathogens transmitted to humans by tick bites, together with the Lyme disease spirochetes, may be involved in the enhancement of the pathological inflammatory response, and consequently, in its possible persistence. However, there are scant clinical data on the possible effects of coinfection on the course of Lyme disease [29].

*A. phagocytophilum* (formerly *Ehrlichia phagocytophila*) is an intracellular, Gram-negative, rickettsia-like bacterium. It is the causative agent of human granulocytic anaplasmosis (HGA). Although this species has long been known to be an animal pathogen, the first cases of human infection were described in 1994 and 1997 in the United States and Europe, respectively [30,31,32]. The disease is probably significantly underdiagnosed due to nonspecific symptoms such as fever, flu-like symptoms, headache, and myalgia, which usually resolve without treatment. According to the National Institute of Public Health (Czech Republic), only 53 cases of HGA were reported here between 2007 and 2017, although a single seroprevalence study conducted in 2014 detected specific antibodies in 34 of 314 individuals tested [33].

*Bartonella* spp. are facultative intracellular microorganisms that are often classified as emerging pathogens in humans. The most important species pathogenic to humans are *Bartonella henselae*, which is responsible for cat scratch disease, *Bartonella quintana*, which causes trench fever, and *Bartonella baciliformis*, which occurs in endemic areas in South America and causes Carrión disease. Infections of humans are usually vector-borne zoonoses transmitted by blood-sucking insects (fleas, sandflies, lice), but in the case of *B. henselae*, direct transmission from cats to humans seems to dominate. The role of ticks in the transmission of infection is not yet clear. In Europe, *Bartonella* infections are mainly found in immunocompromised individuals. In immunocompetent individuals, the infection is often asymptomatic. The most common symptoms include swelling of the lymph nodes, but bacteremia may also occur, which can damage a number of organs, including the heart, eyes, liver, bones, muscles, soft tissues and central nervous system [34,35,36].

*Babesia microti* is a parasite that infects red blood cells and causes malaria-like diseases in humans. Infection is relatively common in endemic areas in the United States. Transmission to humans occurs through ticks of the genus *Ixodes*. The natural host is primarily the white-footed mouse (*Peromyscus leucopus*). However, the infection can also be transmitted through blood transfusions. In immunocompetent individuals, the infection is usually asymptomatic and parasite levels are very low. However, in immunocompromised individuals, especially asplenics, the disease can be life-threatening [37,38]. Thus far, symptomatic infections have only been reported very rarely in Europe, and are mostly imported from the United States.

The aim of the present study was to evaluate the seroprevalence of IgG antibodies against the pathogens that might occur as co-infections in patients with suspected PTLDS in the Czech Republic (*A. phagocytophilum*, *B. henselae*/*quintana* and *B. microti*) and to investigate the possibility of their contribution to the pathogenesis of post-treatment Lyme disease syndrome.

## 2. Materials and Methods

### 2.1. Characteristics of the Studied Patient Group

A total of 103 patients with clinical symptoms that persisted after antibiotic treatment of diagnosed Lyme disease were included in this study. Patients who met the PTLDS criteria (documented disease and antibiotic treatment, long-standing health problems) were selected by the medical staff of the participating clinical departments. From this group, patients with currently positive anti-*Borrelia* IgG antibody blot results were selected for the study. Blood samples were collected by trained medical staff from the Department of Allergology and Clinical Immunology, the Third Department of Internal Medicine-Nephrology, Rheumatology and Endocrinology, and the Department of Immunology of Olomouc University Hospital, Olomouc Region, or provided by the Laboratory of Medical Parasitology and Zoology of the Institute of Public Health Ostrava, Moravian-Silesian Region, Czech Republic. The samples were collected from 2019 to 2021. The basic parameters of the studied population are listed in Table 1. The protocol of the study, including the informed consent of the patients, was approved by the Ethics Committee of the Olomouc University Hospital (reference number 102/18 of June 2018).

### 2.2. Laboratory Tests

Positivity of IgG antibodies against *Borrelia* was confirmed using the Anti-*Borrelia* EUROLINE-RN-AT (EUROIMMUN, Lübeck, Germany) blot diagnostic kit with evaluation by a flatbed scanner and software EUROLineScan Software 3.4 (EU-ROIMMUN, Lübeck, Germany).

Antibodies against *A. phagocytophilum* were determined using the BLOTLINE Anaplasma IgG kit (TestLine Clinical Diagnostics, Brno, Czech Republic) based on strips containing the recombinant antigens p44, Asp62 and OmpA. The tests were performed according to the manufacturer’s instructions and evaluated using a flatbed scanner and Immunoblot Software 1.8.0 (TestLine Clinical Diagnostics, Brno, Czech Republic).

Antibodies against *Bartonella* were determined using the *Bartonella henselae*/*Bartonella quintana* (IgG) immunofluorescence kit (EUROIMMUN, Lübeck, Germany). Serum samples were tested at 320-fold dilution according to the manufacturer’s recommendations. At the last wash, the glasses were counterstained with Evans blue solution. Results were analyzed independently by two trained individuals using an Axioskop fluorescence microscope (Carl Zeiss, Jena, Germany) at 400× magnification and an Olympus DP70 digital camera (Olympus, Tokyo, Japan). Samples with the appropriate pattern but significantly lower intensity than the positive control were classified as borderline. Samples with a fluorescence pattern that differed from that of the positive control were classified as non-evaluable. The positive and negative controls included in the kit were used in each sample series. Due to the high cross-reactivity between *B. henselae* and *B. quintana*, the results in this analysis are reported together as antibodies to *Bartonella* spp.

Antibodies against *Babesia microti* were determined using the Babesia microti IFA IgG antibody kit (Fuller Laboratories, Fullerton, CA, USA). Patient sera were diluted 64-fold and evaluated as described above.

Sera from patients found to have autoantibodies in an immunofluorescence assay were tested for reactivity with individual autoantigens using the immunoblot kits EUROLINE Autoimmune Inflammatory Myopathies 16 Ag (IgG) and EUROLINE ANA Profile 3 plus DSF70 (IgG) (EUROIMMUN, Luebeck, Germany).

All detection kits used are certified for in vitro diagnostics. Data were analyzed using the SPSS v.25 package (IBM Inc., Armonk, NY, USA).

## 3. Results

### 3.1. Detection of Anti-Anaplasma Antibodies

Samples with at least one positive and one borderline reaction of the three *Anaplasma* antigens included in the test were marked as positive by the evaluation software. Of the 103 samples tested, positive results were recorded for 13 samples (12.6%). Samples with 2 borderline or only 1 positive band were classified as equivocal and identified in another 13 samples. The remaining 77 (74.8%) samples were classified as negative.

Of the individual antigens in the positive samples, the OmpA band reacted most frequently. It was positive in 12 (92.3%) and borderline in 1 sample (7.7%). The antigens p44 and Asp62 were consistently positive in 6 samples (46.2%), but Asp62 was borderline confirmed in 3 other samples (23%), as shown in Figure 1. The observed positivity was unevenly distributed by age, with 76.9% of positives detected in individuals over 60 years of age (Figure 2). Using the two-tailed Fisher’s exact test, we found a significant difference in the frequency of positive results between the groups aged 60 years and younger and 61 years and older (*p* < 0.05). The mean age of the positive patients was 64.3 years.

### 3.2. Detection of Anti-Bartonella Antibodies

Antibodies against *Bartonella henselae/Bartonella quintana* were detected by immunofluorescence assay. A positive fluorescence image was observed in 10 samples (9.7%), which matched the pattern and intensity of the positive control included in the test. A similar fluorescence pattern, but with significantly lower intensity of fluorescence and frequency of positive cells than the positive control, was observed in another 14 samples (13.6%). These samples were evaluated as equivocal. Fifty percent of the positive patients were over 60 years of age, with a median of 62.1 years.

An unexpected and interesting finding in a relatively large number of subjects showed a distinct fluorescence pattern not typical of the presence of anti-*Bartonella* antibodies, but corresponding to fluorescence patterns characteristic of autoantibodies. This phenomenon was observed in 12 serum samples (11.6%), with most autoantibody reactivity patterns consistent with antinuclear antibodies (ANA). Antibodies to mitochondria were detected in one sample (Figure 3). These samples were considered unevaluable because the fluorescence of the anti-*Bartonella* antibodies could be masked. The mean age of the autoantibody-positive patients was 64.7 years. The presence of specific autoantibodies was confirmed by diagnostic kits based on the immunoblot method. A positive reaction with at least one specific autoantigen was detected in eight patients, a borderline reaction with at least one specific autoantigen was observed in two patients, and in two patients, we could not detect any reaction with the autoantigens included in the tests used.

### 3.3. Detection of Anti-Babesia Antibodies

None of the samples tested in the immunofluorescence assay showed positive fluorescence comparable to the positive control included in the test kit. All processed samples were therefore scored as negative.

No sample exhibited both anti-*Anaplasma* and anti-*Bartonella* antibodies. Detailed results are published as Appendix A, and a summary is provided in Appendix A.

## 4. Discussion

Chronic problems that persist after antibiotic treatment for Lyme disease are one of the greatest challenges in infectious medicine today. Emerging symptoms such as fatigue, headache, joint and muscle pain or cognitive deficits are often strongly subjectively perceived by patients, but are very difficult to objectify. The problem is the non-specificity and frequency of the mentioned symptoms, especially in the elderly population. The clinical definitions of PTLDS published to date are relatively vague, and the lack of a specific marker makes it very difficult to perform a definitive diagnosis. The pathogenesis of post-treatment complications and the options for therapeutic intervention have long been the subject of intense debate among experts and the public [39]. One of the most controversial issues is the influence of tick-borne coinfections such as anaplasmosis, bartonelosis or babesiosis on the occurrence and/or persistence of these symptoms and the consequent benefit of long-term and/or combined antimicrobial treatment [40,41].

Some studies originating in the United States suggest a high frequency of antibodies to other zoonoses in Lyme disease patients. Krause et al. [29] found anti-*Babesia* antibodies in 40.37% and anti-*Anaplasma* antibodies in 6.83% of patients who tested positive for Lyme disease in their study that focused on tick-borne zoonoses on the northeast coast of the United States. Horowitz and Freeman [42], in their study of 200 patients (193 of whom were from the United States) with PTLDS, reported positive results of serologic or genetic testing for *Anaplasma* infections in 13.5%, *Bartonella* infections in 46.5%, and *Babesia* infections in 52%. However, no details were given on the laboratory methods used in this study; only indirect (antibody titers) or direct (PCR, FISH) tests in different laboratories were mentioned.

In our study, we focused on the determination of antibodies against *A. phagocytophilum*, *B. hansalea*/*quintana* and *B. microti* in patients with suspected PTLDS in order to contribute to understanding of their role in the development of this syndrome in the Central Europe region.

*A. phagocytophilum* is a bacterium with clearly confirmed transmission by ticks of the genus *Ixodes*, pathogenicity to humans, and occurrence in most European countries. In humans, it causes human granulocytic anaplasmosis. The infection clinically manifests as an acute, nonspecific febrile illness with fatigue, headache, muscle, and joint pain. From a clinical standpoint, it is important that the treatment of anaplasmosis be primarily with doxycycline, just as in cases of Lyme disease. In contrast, antibiotics from the penicillin group, cephalosporins, and macrolides (including azithromycin) are ineffective [43,44]. Although symptomatic disease is rarely reported, specific IgG antibodies are found in a relatively high proportion of subjects in Central Europe. In the study conducted at the University Hospital Brno, Czech Republic, *Anaplasma*-specific IgG was found in 3.18%, IgM in 6.05%, and both isotypes in 1.59% of patients with suspected Lyme disease [33]. However, the methodology of the study was based on a combined immunoblot test, which contained only a single *Anaplasma* antigen-p44. In our study, positivity of this antigen was detected in only 46% of the *Anaplasma*-positive results. Kříž et al. [45] compared the frequency of specific IgG in sera of healthy individuals from the general population in the Czech Republic collected in two periods (1978–1989 and 2001). Samples were stored frozen in the serum bank of the National Institute of Public Health in Prague and tested by indirect immunofluorescence. In the first period, 57 of 434 sera (13.1%), and in the second period, 31 of 270 sera (11.5%), tested positive. Although the study population was not selected for a history of tick bites, the frequency of *Anaplasma* IgG positivity in this work is virtually identical to the results of our study in patients with suspected PTLDS (12.6%). A very similar result (11%) was also found in the general adult population in southern Norway [46] and in patients with persistent symptoms attributed to suspected tick bite exposure in Sweden, where *Anaplasma* antibodies were detected in 12% of patients [47]. In a serological study in neighboring Poland, antibodies against *Anaplasma* were found in 11.8% of forest workers and in 9.4% of healthy control blood donors [48]. In Crete, IgG seropositivity was found in 21.4% of blood donors [49], and in a tick-endemic area in Sweden, it was found in as many as 28% of residents [50]. In none of these studies were participants selected on the basis of a history of tick bites or long-term health complications.

Thus, the frequency of anti-*Anaplasma* IgG antibodies found in our study in individuals with suspected PTLDS does not exceed the frequency found in the general population in different areas of Europe with the presence of *Ixodes ricinus* ticks. However, the validity of this comparison is limited by the different methods used. The studies cited above were mainly based on the indirect immunofluorescence method, whereas our study used an immunoblot method based on strips coated with three specific recombinant *A. phagocytophillum* antigens (p44, OmpA and Asp62), which were then scored using a digital system. This method enables reproducible quantification of the intensity of the reaction with the specific antigens by optical densitometry, thus excluding possible subjective factors that may occur in the evaluation of immunofluorescence assays. The use of the immunoblot method also reduces the risk of a false positive reaction caused by antibodies to non-specific antigenic structures. The test kit we used from a local manufacturer is probably the only commercially available immunoblot-based test with multiple recombinant antigens that is certified for in vitro diagnostics.

We observed a significantly unequal distribution of anti-*Anaplasma* antibodies among age groups. More than three-quarters (76.9%) of the positive results occurred in individuals older than the mean and median age in the study group. A similar trend was described in an earlier Czech study, in which the authors described the highest incidence of these antibodies in the age group 60–69 years (16.9%), followed by a group aged 70 years and older (15.8%). The lowest percentage of positive results was in a group of 29 years and younger (3.25%) [33]. In our study, we found positivity in 19% of subjects in the 61–70-year-old category and in 20% of subjects in the 71 years and elder category, but only in 5.8% of subjects younger than 59 years. A possible explanation could be the higher susceptibility of elder people to infection reaching a stage that activates specific immunity, whereas in younger people, non-specific immune mechanisms are more often sufficient to eliminate *Anaplasma* bacteria. However, further studies with a different design would be needed to confirm this hypothesis.

The actual incidence of *Bartonella* infections in humans in Europe remains unclear. From 2012 to 2020, only 76 cases of *B**artonella* infection were reported to the Czech Infectious Diseases Information System (ISIN). However, serological studies from some neighboring countries have found a high frequency of antibodies against *Bartonella* in the population. In a study from Poland, seroprevalence in healthy blood donors was compared with that in rheumatoid patients with unexplained musculoskeletal symptoms. Overall, 23% of the samples tested were positive for antibodies, but no significantly higher frequency was found in rheumatoid patients, in persons previously scratched by a cat, or in persons with a history of tick bites [51]. Different results were obtained in an earlier study from Poland, where persons at increased risk for tick bites (farmers, forestry workers) were seropositive in 30.4% of cases, but the control group was “only” 8.9% [52]. In a study from Germany, positive antibodies against *B. henselae* with titers of 64 and higher were found in 30% of 270 clinically healthy students (age range 21 to 30 years). No significant difference was found between cat breeders and individuals without contact with cats [53]. In a study from the Maryland–Washington D.C. region, USA, antibodies against *Bartonella* spp. were found in 62% of patients examined by a rheumatologist, and positive PCR for these bacteria was found in 41.1% of patients [54].

Due to the high frequency of detection of anti-*Bartonella* antibodies in the general population, the specificity of the tests used is also a topic of debate. Previous studies have indicated serological cross-reactivity between *Bartonella* sp. and pathogenic bacteria of the genera *Chlamydia* and *Coxiella* [55,56]. Vermeulen et al. [57] reported the sensitivity of various tests in the range of 50% to 98% and specificity between 69% and 96%. Some of the tests investigated in this study were also found to be false-positive in patients with cytomegalovirus and Epstein–Barr virus infection. For the commercial test used in our study, the authors of the comparison reported a specificity of 89%. Theoretically, this means that with 103 subjects in our study, 11 to 12 false positives would be expected even in a negative population. However, this is an even higher number than the 10 positive samples we found. The interpretation of the weak positive results that we classified as equivocal is uncertain. If we were to accept them as positive, we would reach 23.3%, which is still consistent with the values reported for the general population in the papers cited above.

In addition, comparisons of results between studies are complicated by the fact that the cut-off point for seropositivity is not uniformly accepted. Although the manufacturer of the kit we used recommends that the serum tested be diluted 320-fold, another immunofluorescence kit recommends that the serum be evaluated with only a 64-fold dilution [58]. A major advantage would be the introduction of new serological methods based only on reactions with specific antigens, such as immunoblotting, which could be used to confirm the reactive results of indirect immunofluorescence screening. Similar two-step tests, in this case, the combination of ELISA and blot, have long been an accepted standard for the serological diagnosis of Lyme disease.

Another unresolved issue is also the possibility of *Bartonella* sp. transmission by the vectors of *B. burgdorferi* sensu lato spirochetes, hard ticks from the *I. ricinus* complex. Infection of *Ixodes* ticks with *Bartonella* by an artificial membrane feeding technique, resulting in transovarial and trans-stadial transmission, has previously been demonstrated [59,60]. In one of these studies, the transmission of *Bartonella* via a tick to a cat was successful, but the power of the study is somewhat limited by the artificial infection of the ticks used with a large number of laboratory-adapted bacterial strains [59]. The detection of DNA of *Bartonella* sp. in ticks has also repeatedly been described in Europe [61,62,63]. In 2001, Eskow et al. [64] published a series of cases with presumed tick-borne concurrent neurological infection by *Borrelia burgdorferi* and *Bartonella henselae*. However, concerns have been raised about the adequacy of the conclusions drawn from the results presented and the methodology used in the study itself, including the specificity of the primers used in the PCR analyses [65,66].

Nevertheless, there is no convincing evidence of the competence of ticks to transmit *Bartonella* species between domestic or wild animals and humans [40,66]. However, symptomatic infection by the transmission of *B. hensalae* directly from an infected mammal, usually a cat, has been well described [67]. The cause of seropositivity in asymptomatic individuals who have not been scratched by a cat remains unclear.

In the United States, *Babesia microti* is endemic in the Northeast and upper Midwest regions and is one of the most common tick-borne pathogens [68]. According to the CDC, approximately 2000 human cases of babesiosis are reported annually. To date, more than 24,000 cases have been identified in the United States, and more than 1000 in neighboring Canada [69]. In Europe, clinical cases of human infection are rarely reported, although *B. microti* has been found in *Ixodes* ticks in many European countries [70,71]. Overall, only about 50 symptomatic *Babesia* infections have been reported in Europe, but most of these have been attributed to another *Babesia* species, particularly *B. divergens*, which mainly affects immunocompromised individuals after splenectomy [71]. The natural hosts of this parasite are mainly cattle, which are not a reservoir for *B. burgdorferi* sensu lato. To date, only one case of *B. microti* infection has been described in the Czech Republic. The patient was an entomologist returning from a business trip during which he had collected ticks in Connecticut, USA. The disease manifested as fever, fatigue, loss of appetite and dark urine. Laboratory tests showed severe anemia (hemoglobin 65 g/L), leukopenia, trombocytopenia, and elevated levels of C-reactive protein (98 g/L). The presence of the parasite in the erythrocytes was subsequently detected microscopically, but with a positivity of only 0.41% of the erythrocytes. At the same time, anaplasmosis was diagnosed based on serology and he was treated with a combination of antiparasitics and antibiotics, resulting in the remission of symptoms [72].

Serological studies show a low percentage of positive antibodies against *B. microti* in persons with a history of tick bites, or even in the general population in some European localities. In Belgium, seropositivity of anti-*B. microti* antibodies was observed in 18 of 199 (9%) patients with clinical symptoms after a tick bite [73]. Conflicting results have been provided by two recent studies from Sweden. Svensson et al. [74] found anti-*Babesia* IgG antibodies in 16.3% of *Borrelia*-positive individuals and in 2.5% of healthy controls, whereas Nillson et al. [47] did not detect these antibodies in any of the 224 patients with persistent symptoms due to suspected tick bite exposure in their comprehensive study. The reason for this difference could be the natural endemicity of the presence of the parasite, in addition to the different methods used. In our study, we did not find a convincing positive result in any of the samples tested.

An unexpected and valuable finding of our study was the detection of autoantibodies in 12 (11.6%) patients by immunofluorescence assay using human infected cell culture to determine antibodies against *Bartonella*. In most cases, these were different patterns belonging to the antinuclear antibody group; in one sample, antibodies with a fluorescence typical of antimitochondrial antibodies were found. We confirmed the presence of autoantibodies in eight of these samples by immunoblotting. A borderline result was obtained in two samples and a negative result in the other two samples. However, the negativity of the blots does not exclude reactivity with other autoantigens absent from the assays used.

Although causality with prior infection cannot be clearly established, the results are consistent with a number of previous studies in patients with complicated forms of Lyme disease and post-Borreliosis syndrome, in which antibodies to a variety of autoantigens were found, such as annexin A2 [75], apolipoprotein B-100 [76], endothelial cell growth factor [77], matrix metalloproteinase-10 [78], phospholipids [79], cyclophilin [80], neural tissue [81,82], γ-enolase [83], ANA, and myositis-associated antibodies [47]. The wide range of specificities of the autoantibodies found suggests the non-specific polyclonal activation of cells of the immune system rather than the cross-reactivity of microbial antigens with autoantigens. In any case, these results must be interpreted with caution, because the presence of autoantibodies does not necessarily indicate the presence of a specific autoimmune disease. However, autoantibodies may be a sign of activation of the immune system, whether long-term after infection or due to the persistence of immunogenic material in the organism.

The limitations of our study are mainly determined by the general limitations of the serological methods used. The co-prevalence of antibodies alone cannot distinguish between a true co-infection and two independent infections separated in time. To determine the frequency of co-infection itself, it would be appropriate to perform direct pathogen detection (PCR, microscopy, and cultivation) in diagnosed patients before starting antibiotic treatment. However, in clinical practice, this is not performed, mainly because of the cost and limited availability of these methods to primary care physicians, where most cases of Lyme disease are diagnosed. The role of co-infections in the pathogenesis of PTLDS would be indicated by a higher frequency of seropositivity against other selected tick-borne infections in this group of chronic patients than in the general population, which was not observed in our study.

## 5. Conclusions

In our study, we found positive levels of IgG antibodies against *A. phagocytophilum* in 12.6% of samples from patients with suspected PTLDS from the Olomouc and Moravian-Silesian regions, Czech Republic, and against *Bartonella henselae*/*Bartonnela quintana* in 9.7% of samples. Positivity was detected in only a relatively small proportion of samples, and the frequency observed did not exceed the seropositivity found in previous studies in populations without health problems with suspected post-Borreliosis etiology. For antibodies against *A. phagocytophilum*, we observed a significant dependence of positivity on patient age.

We assume that the results of our study do not show a significant contribution of the co-infections studied above to the pathogenesis of PTLDS in Central Europe and the Czech Republic. However, the positivity of specific antibodies, especially against the bacterium *A. phagocytophilum*, still deserves medical attention. The benefit of treating the symptoms associated with tick bites with tetracycline antibiotics, which are also effective against the above-mentioned infections, is emphasized.

An important finding was the presence of autoantibodies in 12 (11.6%) of the patients detected by immunofluorescence assay using human infected cell culture to determine antibodies to *Bartonella.* The presence of autoantibodies was confirmed in eight samples by immunoblotting. An association between previous *Borrelia* infection, the presence of autoantibodies and clinical symptoms could not be clearly demonstrated in our study, and further studies with sufficiently large control groups are needed to clarify the possible association.

## Figures and Tables

**Figure 1 microorganisms-09-02217-f001:**
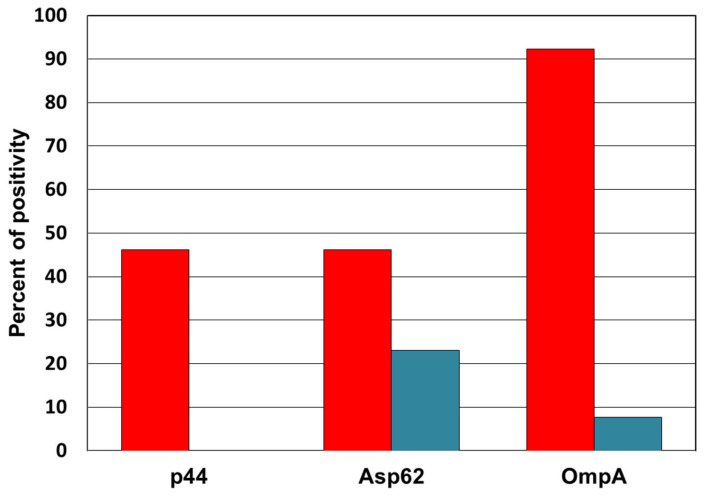
Representation of positive (red) and borderline (blue) reactions of individual antigens in samples with a positive result in the anti-*Anaplasma* antibody test.

**Figure 2 microorganisms-09-02217-f002:**
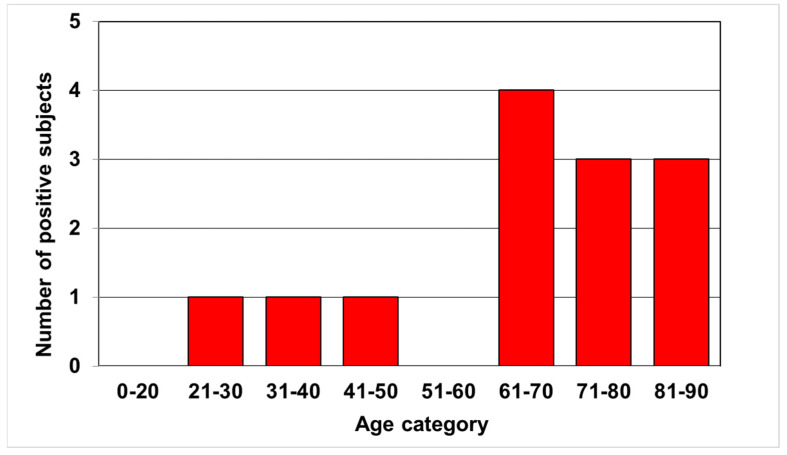
Distribution of positive anti-*Anaplasma* antibody results by age. The difference in the frequency of positive results between the groups of 60 years and younger and 61 years and older was found to be statistically significant (two-tailed Fisher’s exact test, *p* <0.05).

**Figure 3 microorganisms-09-02217-f003:**
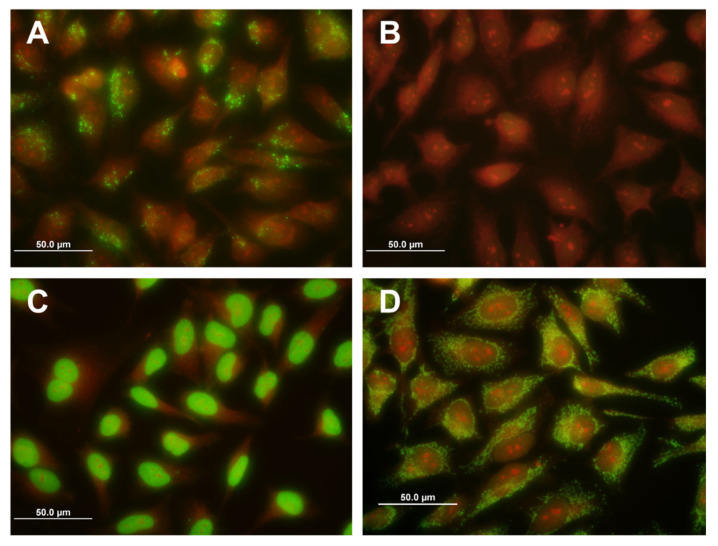
Fluorescence patterns observed in the study. (**A**) Positive reaction to *Bartonella* antibodies, (**B**) negative reaction, (**C**) antinuclear antibodies (ANAs), (**D**) antimitochondrial antibodies (AMA). Images were created using the software assembly of images in green and red fluorescence channels at a total magnification of 400×.

**Table 1 microorganisms-09-02217-t001:** Characteristics of studied patient group.

Number of Subject	103
Average Age (min.–max.)	57.4 (6–90) years
Median of Age	59 years
Females/Males	46/57
0–20 years	3
21–30 years	8
31–40 years	6
41–50 years	19
51–60 years	16
61–70 years	21
71–80 years	23
81+ years	7

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
