# Peer review of "Seroprevalence of Antibodies against Tick-Borne Pathogens in Czech Patients with Suspected Post-Treatment Lyme Disease Syndrome"

_microorganisms, 2021, doi:10.3390/microorganisms9112217_

Round 1

Reviewer 1 Report

Review report

Manuscript ID: microorganisms-1405741
Type of manuscript: Article
Title: Seroprevalence of antibodies against tick-borne pathogens in Czech patients with suspected post-treatment Lyme disease syndrome.

Authors: Kristýna Sloupenská, Jana Doležílková, Barbora Koubková, Beáta Hutyrová, Mojmír Račanský, Pavel Horak, Maryna Golovchenko, Milan Raska, Natalie Rudenko, Michal Křupka
Section: Parasitology

Special Issue: Borrelia and Lyme Disease

Since Lyme disease (Lyme borreliosis), was identified in the 1970s, its importance for public health has become clear, and it is the most common tick-borne disease of humans in the Northern Hemisphere. Lyme disease is caused by the Borrelia burgdorferi (sensu lato) complex of spirochete bacteria, comprising at least 22 genospecies globally. Borrelia genospecies vary in pathogenicity and cause different symptoms. They also differ in geographic distribution, Ixodidae vector and transmission host(s). Due to this complexity, there are still a lot of knowledge gaps within its etiology, epidemiology, ecology and clinic. One of the points that needs better understanding is the hypothesised importance of co-infections in the pathogenesis of post-treatment Lyme disease syndrome (previously known as chronic Lyme disease).

Broad comments: 

The aim of this study was to investigate the importance of coinfeccions in the pathogenesis of the post-treatment Lyme disease syndrome (PTLDS); specifically the authors evaluated the seroprevalence of IgG antibodies against pathogens that might occur as co-infections in patients with suspected PTLDS (A. phagocytophillum, B. henselae/quintana and B. microti).

The authors have adressed important points: the complexity of Lyme disease and PTLDS diagnoses and the possible involvement of co-infections in the pathogenesis of PTLDS. These are in fact current points of debate in the scientific and medical communities and, of high importance to the general public.

The text is very well written and clear; the authors establish a good link between all the sections. Both the introduction and the discussion show a good knowledge about the topic.

Although the authors have compared the results of individuals with possible PTLDS with results of healthy individuals from previous papers, a good addition to this paper was to have a case-control study, with the same methods applied to a group of PTLDS and a healthy group.

Minor changes:

Abstract:

Lines 22 (againstthe )and 26 (notconfirmed) – separate the words.

Introduction:

Lines 60-62: A suggestion is to include how the prevention is based – Its prevention is based on minimising the risk of exposure to ticks and tick bites, rapid removal of attached ticks and early recognition of symptoms to initiate prompt treatment.  

Materials and methods:

2.1. Study population

The subtitle is not adequate. Study population is a subset of the target population (i.e., population with PTLDS in Central Europe) from which the sample is actually selected. The study population seems to be the population in Czech Republic with PTLDS – what you have is a sample of 103 individuals with those characteristics. Change the subtitle or define study population in the text.

This section lacks information of how the samples were selected. Such as: are these 103 patients the total of patients in Czech Republic with PTLDS? If not, how the sample was selected? Random sampling? Just based on gender and age? Based on their postcode or if they lived in an urban or rural environment? Explain the criteria for this selection.

Overall, there was a similar number of males and females, but with an uneven distribution within the different age groups. It would have been good to compare the results of this group with the results of a control group (healthy group of the same age groups/gender).

Change accordingly the title of table 1.

Discussion:

Line 248 –“ explanation of their role in the pathogenesis of this syndrome in Central Europe”.

Line 267 – “out of 434 sera (13.1%) and in the second period 31 out of 270 sera (11.5%) tested positive”.

Line 273 - were detected in 12% of the patients [38]

Have any of the studies in healthy population you have referred, compared the frequency of antibodies against the three pathogens by age group?

Lines 281-285 – It would be good that you had a control group of healthy individuals, of the same age groups, to test the differences found.

Lines 285-287 – could you explain a bit better the value of the method you used?

Lines 318-319 – “another immunofluorescence kit recommends evaluating the serum only with a 64-fold dilution. “ Could you give a ref?

Author Response

Dear Reviewer,

We are grateful for your help in improving of our manuscript. Your comments, edits and advices are greatly appreciated, thank you. We addressed all of them in our best beliefs. Our responses are below and we hope you will consider them to be sufficient.

Minor changes:

Abstract:

Lines 22 (againstthe )and 26 (notconfirmed) – separate the words.

Corrected, thank you.

Introduction:

Lines 60-62: A suggestion is to include how the prevention is based – Its prevention is based on minimising the risk of exposure to ticks and tick bites, rapid removal of attached ticks and early recognition of symptoms to initiate prompt treatment. 

The following sentence has been added to the Introduction:
The best way to prevent infection thus remains minimizing of tick bite risk (suitable clothing with long sleeves and trousers and the use of repellent during outdoor activities) and the rapid removal of ticks that have already been attached. Early recognition of symptoms and use of appropriate antibiotics is essential for the success of treatment.

Materials and methods:

2.1. Study population

The subtitle is not adequate. Study population is a subset of the target population (i.e., population with PTLDS in Central Europe) from which the sample is actually selected. The study population seems to be the population in Czech Republic with PTLDS – what you have is a sample of 103 individuals with those characteristics. Change the subtitle or define study population in the text.

Subtitle has been changed to „Characteristics of studied patients group“, thank you.

This section lacks information of how the samples were selected. Such as: are these 103 patients the total of patients in Czech Republic with PTLDS? If not, how the sample was selected? Random sampling? Just based on gender and age? Based on their postcode or if they lived in an urban or rural environment? Explain the criteria for this selection.

Information about the samples selection was added and the description of the methodology in the manuscript has been specified.

Patients were selected by clinics on the basis of meeting the criteria for PTLDS - a documented experience of antibiotic treatment of Lyme borreliosis and persistent problems despite previous treatment. Eligible patients meeting the criteria were selected by the clinicians of the cooperating departments between 2019 and 2021. Samples of selected patients, in which the presence of specific IgG antibodies was confirmed by western blot were processed.

Overall, there was a similar number of males and females, but with an uneven distribution within the different age groups. It would have been good to compare the results of this group with the results of a control group (healthy group of the same age groups/gender).

We agree that the informative value of the study would be increased by the inclusion of a control group. Unfortunately, this extension is not realistic in the timeframe of the revision of the manuscript and would require additional funding. On the other hand, the literature data from previously published studies in the general population in Central Europe are rather extensive and, in our opinion, sufficient for comparison.

In the future, we plan to expand our study in terms of the number of monitored patients as well as the number of monitored spectrum of antibodies against specific pathogens. We also plan to compare PTLDS group with healthy age-matched controls and disease control group based on samples from patients with chronic fatigue syndrome/myalgic encephalomyelitis negative for anti-Borrelia antibodies.

Change accordingly the title of table 1.

Title has been changed, thank you.

Discussion:

Line 248 –“ explanation of their role in the pathogenesis of this syndrome in Central Europe”.

Corrected, thank you.

Line 267 – “out of 434 sera (13.1%) and in the second period 31 out of 270 sera (11.5%) tested positive”.

Corrected, thank you.

Line 273 - were detected in 12% of the patients [38]

Corrected, thank you.

Have any of the studies in healthy population you have referred, compared the frequency of antibodies against the three pathogens by age group?

The increased incidence of antibodies against Anaplasma phagocytophillum in the elderly patients has been described also in a paper written and published in Czech, cited in another part of the discussion: Ref. 33: Dvořáková Heroldová, M.; Dvořáčková, M. [Seroprevalence of Anaplasma phagocytophilum in patients with suspected Lyme borreliosis] In Czech. Epidemiol. Mikrobiol. Imunol. 2014, 63, 297–302.

 A paragraph focused on this phenomenon has been added to the discussion.

Lines 281-285 – It would be good that you had a control group of healthy individuals, of the same age groups, to test the differences found.

We would like to refer you to one of the previous comments on the control group in the section on materials and methods above. Thank you.

Lines 285-287 – could you explain a bit better the value of the method you used?

The description of principles and benefits of method over immunofluorescence has been extended in the manuscript:

The studies cited above were mainly based on the indirect immunofluorescence method, whereas our study used an immunoblot technique based on strips lined with 3 specific recombinant A. phagocytophillum antigens (p44, OmpA and Asp62) followed by evaluation with a digital system. This method enables reproducible quantification of the serum reaction intensity with the specific antigens by means of optical densitometry and so exclude possible subjective factors that may occur in the evaluation of immunofluorescence tests. The use of the immunoblot method also reduces the risk of a false positive reaction caused by antibodies against non-specific antigenic structures. The test set of a local producer used by us is probably the only commercially available immunoblot-based test with several recombinant antigens certified for in vitro diagnostics.

Lines 318-319 – “another immunofluorescence kit recommends evaluating the serum only with a 64-fold dilution. “ Could you give a ref?

The corresponding reference has been added to the manuscript:

Ref. 58: Maurin, M.; Rolain, J.M.; Raoult, D.Comparison of In-House and Commercial Slides for Detection by Immunofluorescence of Immunoglobulins G and M against Bartonella henselae and Bartonella quintana. Clin Diagn Lab Immunol. 2002, 9, 1004-9, doi: 10.1128/cdli.9.5.1004-1009.2002.

Reviewer 2 Report

Sloupenska et al. report results from a seroprevalence study conducted in the Czech Republic. Serum samples were collected from 103 patients with clinical symptoms that persisted after treatment for Lyme disease. Patients had a positive anti-Borrelia IgG western blot at the time of enrollment in the study. Serum was tested for antibodies against Anaplasma phagocytophilum, Bartonella henselae/quintana, and Babesis microti.

The researchers report seroprevalences of 12.6%, 9.7%, and 0% for A. phagocyophilum, Bartonella spp., and B. microti, respectively. These are similar to prevalences reported for the pathogens in other studies conducted in the region. Therefore, the researchers conclude there is poor support for the hypothesis that co-infections with other tick-borne pathogens contribute to Post-Treatment Lyme Disease Syndrome (PTLDS). They detect autoantibodies in 11.6% of their samples, suggesting to them that their results were more supportive of the alternate hypothesis that PTLDS may be associated with an autoimmune sensitization disorder.

Overall, there are limited results reported in the manuscript. The sample size is small and not much information is provided about the people in the sample, besides age. Therefore, the impact of this work is very limited. The meaning of their findings would have been improved if they would have included a group of Lyme disease patients without continuing symptoms, as a control group, in the study. Without the control group, it is hard to know if the percentages of elevated titers observed has anything to do with the PTLDS status of the group. Even the 11.6% of positive autoantibodies detected, though reported to be unexpected and to support the autoimmune hypothesis for PTLDS, can’t be evaluated in the context of Lyme disease or PTLDS, as we don’t know what the percentage of autoantibodies may have been in a control group. Seropositivity for autoantibodies is common in the general population (see Haller-Kikkatalo et al. 2017 Scientific Reports 7, article number 44846), and there are cautions against interpreting much from a single elevated ANA test (e.g., https://www.uptodate.com/contents/antinuclear-antibodies-ana-beyond-the-basics).  

For this reason, I am unable to recommend publication of the manuscript.

Other minor comments:

Abstract, lines 26-30, would suggest changing, “An unexpected finding of our study…” to something like, “We did, however, detect the presence of autoantibodies, mostly antinuclear, in 11.6%..., which is in agreement with previously published studies. This suggests our results are in agreement with …”

Line 154, phagocytophilum misspelled.

Lines 187-189, researchers state they had insufficient numbers to assess significance of the age difference they detected in the distribution of positive antibodies for A. phagocytophilum. I think they would be able to do a 2 x 2 contingency test using Fisher’s exact test to determine significance, for small samples sizes. Hence, they could compare the numbers of positive and negative tests in the 2 age groups, < 60 years old and ≥ 60 years old.

Author Response

Dear Reviewer,

Thank you for your time, your thoughts and comments that we addressed in our best beliefs.

We agree that the informative value of the study would be increased by the inclusion of a control group. Unfortunately, this extension is not realistic in the timeframe of the revision of the manuscript and would require additional funding. On the other hand, the data from earlier published studies on the general population in Central Europe are rather extensive and, in our opinion, sufficient to be used for comparison.

We also fully agree that the results of autoantibody tests need to be interpreted with caution, that is why we have expanded the corresponding part in the discussion. However, we believe that our results on autoantibody detection are significant enough to be mentioned and discussed in the publication, particularly from a methodological point. The immunofluorescence method for detecting antibodies  against Bartonella sp. in the presence of autoantibodies is not evaluable, but its presence in previous studies based on the same method is not mentioned or discussed.

Other minor comments:

Abstract, lines 26-30, would suggest changing, “An unexpected finding of our study…” to something like, “We did, however, detect the presence of autoantibodies, mostly antinuclear, in 11.6%..., which is in agreement with previously published studies. This suggests our results are in agreement with …”

Sentence has been modified, thank you

Line 154, phagocytophilum misspelled.

The typo has been corrected, thank you.

Lines 187-189, researchers state they had insufficient numbers to assess significance of the age difference they detected in the distribution of positive antibodies for A. phagocytophilum. I think they would be able to do a 2 x 2 contingency test using Fisher’s exact test to determine significance, for small samples sizes. Hence, they could compare the numbers of positive and negative tests in the 2 age groups, < 60 years old and ≥ 60 years old.

The statistical analysis was added to the results:

Age distribution of A. phagocytophilum-specific antibodies positivity.

Age

Negative

Positive

Total

0-60

49

3

52

61-99

41

10

51

Total

90

13

103

The association among subjects age and A. phagocytophilum-specific antibodies is considered to be statistically significant. Subjects were separated into groups 0-60 and 61-99 years. The two-tailed Fisher's exact test, P value equals 0.0413.

Reviewer 3 Report

Line 50: A bull's eye rash, when it occurs is often regarded as pathognomonic; however amongst all rashes which may occur in Lyme disease, it occurs in a minority.  So it is inaccurate to say it is the most 'common' clinical manifestation of early Lyme disease.

Lines 58-59: "Long-term antibiotic.....not recommended" does not take in to account respectable minority view well-supported by ample peer-reviewed publications. Failure to discuss this legitimate controversy within this field is a deficiency.

Line 68-69: Post-treatment Lyme disease is a research concept and is a limited subset of chronic Lyme disease, circumscribed so as to be readily amenable to research studies. Failure to cite Shor et. al.ILADS Evidenced Based Definition of Chronic Lyme disease connotes uncritical bias and/or unfamiliarity with the extensive literature describing chronic Lyme due to chronic infection with or without prior antibiotic treatment. Rather than making dogmatic absolutist statements, discussion of a diversity of viewpoints in this field would be more balanced.

Lines 85-89: Once again, a political statement rather than a dispassionate presentation of data.  Both co-infection(s) and possible chronic persistent borrelial infection can add to the complexity of the clinical picture and require an open-minded approach to how to improve patients' condition

Lines 203-210: What features of the test-kit for assessing anti-bartonella antibodies made it suitable for interpreting auto-immune markers of differing characters? Is this a legitimate inference?  Why use a test-kit designed for anti-bartonella antibodies to gauge auto-immune markers? Why were not standard methods used to test for auto-immunity using standardized test-kits or methods for those disorders?  If it is legitimate to adapt the anti-bartonella antibody test-kit for measuring auto-antibodies, discussion of this with an explication justifying this is appropriate. If it cannot be justified, then the sera, if still available, ought to be subjected to auto-immune testing using standard methodologies.

Lines: 288-301: Consideration of inclusion of the following references in the discussion might seem pertinent.

Maggi RG, Mozayeni BR, Pultorak EL, Hegarty BC, Bradley JM, Correa M, Breitschwerdt EB. Bartonella spp. bacteremia and rheumatic symptoms in patients from Lyme disease-endemic region. Emerg Infect Dis. 2012 May;18(5):783-91. doi: 10.3201/eid1805.111366. PMID: 22516098; PMCID: PMC3358077.

Mozayeni BR, Maggi RG, Bradley JM, Breitschwerdt EB. Rheumatological presentation of Bartonella koehlerae and Bartonella henselae bacteremias: A case report. Medicine (Baltimore). 2018 Apr;97(17):e0465. doi: 10.1097/MD.0000000000010465. PMID: 29703000; PMCID: PMC5944489.

Lines 373-388: Explication of adapting the bartonella antibody test-kit to inferences of auto-immunity, again would seem appropriate. Surely, markers of auto-immunity can be seen in chronic infections including Lyme disease and others. So, in a genetically predisposed host, this may be 'triggered' by the infection in the absence of ongoing infection, but also, chronic persistent infection can initiate and maintain this; and sometimes auto-immune markers diminish or revert to negative with application of antibiotics.  Fuller discussion of this possibility, with more review of the pertinent literature would enhance the manuscript.

There ought to be a section discussing the limitations of the study. Not included were any direct detection methods for either Lyme disease or any of the co-infections studied.  Also, not taken in to account is the diversity of bartonella strains that can infect humans, beyond merely B. henselae/quintana.  

There are novel observations here based on original research findings and these are worthy of publication.  However, more dispassionate and balanced discussion of the science would improve the mss.

Author Response

Dear Reviewer,

Thank you so much for your comments and advices that we tried to address properly.

First of all we would like to say, that we agree with your remarks that some of our statements were rather strict and in some places were lacking the objective evaluation. It was not done on purpose, we appologise for this. We did our best to expand the discussion of controversial questions, added more citations that reflect different opinions as well as highlight the complexity of the discussed subject. Innapropriate statements were either removed or re-written, using the already published studies.

We are grateful for your critique and truly hope that addressing it we improved our manuscript significantly.

Our detailed responces are below:

Line 50: A bull's eye rash, when it occurs is often regarded as pathognomonic; however amongst all rashes which may occur in Lyme disease, it occurs in a minority.  So it is inaccurate to say it is the most 'common' clinical manifestation of early Lyme disease.

Mentioned abstract was modified significantly, thank you for your advice.

However, we would like to say that this sentence is based on the generally reported occurrence of erythema migrans – WHO (https://www.euro.who.int/__data/assets/pdf_file/0008/246167/Fact-sheet-Lyme-borreliosis-Eng.pdf),  CDC https://www.cdc.gov/lyme/signs_symptoms/index.html ), and eCDC (https://www.ecdc.europa.eu/sites/default/files/media/en/healthtopics/emerging_and_vector-borne_diseases/tick_borne_diseases/public_health_measures/Documents/HCP_factsheet_LB_highres.pdf).  However, we recognize that the rash at the site of tick attachment does not always have a typical target or „bull´s eye“  morphology. Some sources report this morphology less often in the United States than in Europe, where this work originated (e.g. . Marques  et al., Emerg Infect Dis. 2021;27(8):2017-2024).

Lines 58-59: "Long-term antibiotic.....not recommended" does not take in to account respectable minority view well-supported by ample peer-reviewed publications. Failure to discuss this legitimate controversy within this field is a deficiency.

We re-wrote this passage and extended the discussion accoring to your recommendation, thank you.

Line 68-69: Post-treatment Lyme disease is a research concept and is a limited subset of chronic Lyme disease, circumscribed so as to be readily amenable to research studies. Failure to cite Shor et. al.ILADS Evidenced Based Definition of Chronic Lyme disease connotes uncritical bias and/or unfamiliarity with the extensive literature describing chronic Lyme due to chronic infection with or without prior antibiotic treatment. Rather than making dogmatic absolutist statements, discussion of a diversity of viewpoints in this field would be more balanced.

We re-wrote this passage and extended the discussion accoring to your recommendation. Mentioned citation was added as well as others, that represent different opinion. We agree that discussion of diversity of viewpoints was on place. Thank you.

Lines 85-89: Once again, a political statement rather than a dispassionate presentation of data.  Both co-infection(s) and possible chronic persistent borrelial infection can add to the complexity of the clinical picture and require an open-minded approach to how to improve patients' condition

The passage was re-written and extended accoring to your recommendation, thank you.

Lines 203-210: What features of the test-kit for assessing anti-bartonella antibodies made it suitable for interpreting auto-immune markers of differing characters? Is this a legitimate inference?  Why use a test-kit designed for anti-bartonella antibodies to gauge auto-immune markers? Why were not standard methods used to test for auto-immunity using standardized test-kits or methods for those disorders?  If it is legitimate to adapt the anti-bartonella antibody test-kit for measuring auto-antibodies, discussion of this with an explication justifying this is appropriate. If it cannot be justified, then the sera, if still available, ought to be subjected to auto-immune testing using standard methodologies.

The test was not intentionally used to detect autoantibodies. Obtained result is essentially an undesirable interaction, making it impossible to evaluate anti-Bartonella antibodies.

The assay used to detect antibodies to Bartonella is based on fixed and permeabilized cell lines of human origin, as well as standard assays for the determination of autoantibodies. These assays differ only in that the cell line intended to detect antibodies to Bartonella was infected with these bacteria in vitro before processing. The possibility of a positive reaction with autoantibodies is also mentioned by the manufacturer of the kit in the manual:  „If a patient samples contains antibodies against Bartonella henselae, the same pattern must essentially be obtained as for the positive control.  If the cell nuclei or the cytoplasm of all cells are stained, i.e. also those of non-infected cells, antinuclear antibodies or antibodies against mitochondria and other cell antigens are present“. The test procedure was thus not adapted in any way. However, we supplemented the examination using the blot technique and inserted its results into the result part of the manuscript.

Lines: 288-301: Consideration of inclusion of the following references in the discussion might seem pertinent.

Maggi RG, Mozayeni BR, Pultorak EL, Hegarty BC, Bradley JM, Correa M, Breitschwerdt EB. Bartonella spp. bacteremia and rheumatic symptoms in patients from Lyme disease-endemic region. Emerg Infect Dis. 2012 May;18(5):783-91. doi: 10.3201/eid1805.111366. PMID: 22516098; PMCID: PMC3358077.

The study has been used for discussion and added to the appropriate part of manuscript.

Mozayeni BR, Maggi RG, Bradley JM, Breitschwerdt EB. Rheumatological presentation of Bartonella koehlerae and Bartonella henselae bacteremias: A case report. Medicine (Baltimore). 2018 Apr;97(17):e0465. doi: 10.1097/MD.0000000000010465. PMID: 29703000; PMCID: PMC5944489.

The citation was added to the paragraph describing possible manifestations of infection in the Introduction part of the manuscript.

Lines 373-388: Explication of adapting the bartonella antibody test-kit to inferences of auto-immunity, again would seem appropriate. Surely, markers of auto-immunity can be seen in chronic infections including Lyme disease and others. So, in a genetically predisposed host, this may be 'triggered' by the infection in the absence of ongoing infection, but also, chronic persistent infection can initiate and maintain this; and sometimes auto-immune markers diminish or revert to negative with application of antibiotics.  Fuller discussion of this possibility, with more review of the pertinent literature would enhance the manuscript.

As described above, the anti-Bartonella antibody assays have not been modified in any way, tests are capable to capture autoantibodies as they are. The interpretation of the occurrence of autoantibodies has been extended in the manuscript. We agree that more attention should be devoted to the careful interpretation of the results. Thank you.

There ought to be a section discussing the limitations of the study. Not included were any direct detection methods for either Lyme disease or any of the co-infections studied.  Also, not taken in to account is the diversity of bartonella strains that can infect humans, beyond merely B. henselae/quintana. 

A paragraph describing the limitations of the study has been added to the manuscript. In the introduction, we state that the study focused only on a few specific pathogens most commonly associated with the possible pathogenesis of PTLDS. We consider the absence of tests certified for human use to be a significant obstacle to the determination of antibodies against other Bartonella species.

There are novel observations here based on original research findings and these are worthy of publication.  However, more dispassionate and balanced discussion of the science would improve the mss.

We did our best to improve the discussion of the manuscript, extended the number of cited published studies and dispassionately presented the different opinions. Thank you.

Round 2

Reviewer 2 Report

I thank the authors for their revisions and understand the are not able to add a control group to the study.  I believe they have written the paper emphasizing the shortcomings of the study and the need for additional work.

Reviewer 3 Report

The Reviewer's concerns have all been satisfactorily addressed and acceptance for publication is recommended.  Unsupportable statements based on dogma have been omitted. Further clarification of methodologies and discussion of limitations have been added.  The mss. is improved.

RE: Line 139, consider "...hopefully vanquishing this disease." (instead of "...hopefully the disabling of this disease.") (Assuming the Reviewer is correctly understanding what the authors are trying to convey).